# Dissecting transcriptomic signatures of neuronal differentiation and maturation using iPSCs

Emily E. Burke[1,11], Joshua G. Chenoweth[1,11], Joo Heon Shin[1], Leonardo Collado-Torres [1], Suel-Kee Kim[1], Nicola Micali[1], Yanhong Wang [1], Carlo Colantuoni[1], Richard E. Straub[1], Daniel J. Hoeppner[1], Huei-Ying Chen[1], Alana Sellers[1], Kamel Shibbani[1], Gregory R. Hamersky[1], Marcelo Diaz Bustamante[1], BaDoi N. Phan [1], William S. Ulrich[1], Cristian Valencia[1], Amritha Jaishankar[1], Amanda J. Price [1,2], Anandita Rajpurohit[1], Stephen A. Semick[1], Roland W. Bürli[3], James C. Barrow[1], Daniel J. Hiler[1], Stephanie C. Page[1], Keri Martinowich[1,4,5], Thomas M. Hyde[1,4,6], Joel E. Kleinman[1,4], Karen F. Berman[7], Jose A. Apud[7], Alan J. Cross[8], Nicholas J. Brandon [8], Daniel R. Weinberger [1,2,4,5,6], Brady J. Maher [1,4,5], Ronald D.G. McKay[1]* & Andrew E. Jaffe [1,2,4,5,9,10]*

Human induced pluripotent stem cells (hiPSCs) are a powerful model of neural differentiation and maturation. We present a hiPSC transcriptomics resource on corticogenesis from 5 iPSC donor and 13 subclonal lines across 9 time points over 5 broad conditions: self-renewal, early neuronal differentiation, neural precursor cells (NPCs), assembled rosettes, and differentiated neuronal cells. We identify widespread changes in the expression of both individual features and global patterns of transcription. We next demonstrate that co-culturing human NPCs with rodent astrocytes results in mutually synergistic maturation, and that cell type-specific expression data can be extracted using only sequencing read alignments without cell sorting. We lastly adapt a previously generated RNA deconvolution approach to single-cell expression data to estimate the relative neuronal maturity of iPSC-derived neuronal cultures and human brain tissue. Using many public datasets, we demonstrate neuronal cultures are maturationally heterogeneous but contain subsets of neurons more mature than previously observed.

[1] Lieber Institute for Brain Development, Baltimore, MD, USA. [2] McKusick Nathans Institute of Genetic Medicine, Johns Hopkins School of Medicine, Baltimore, MD, USA. [3] Neuroscience, IMED Biotech Unit, AstraZeneca, Cambridge, UK. [4] Department of Psychiatry and Behavioral Sciences, Johns Hopkins School of Medicine, Baltimore, MD, USA. [5] Department of Neuroscience, Johns Hopkins School of Medicine, Baltimore, MD, USA. [6] Department of Neurology, Johns Hopkins School of Medicine, Baltimore, MD, USA. [7] Clinical and Translational Neuroscience Branch, NIMH Intramural Research Program, Bethesda, MD, USA. [8] Neuroscience, IMED Biotech Unit, AstraZeneca, Boston, MA, USA. [9] Department of Mental Health, Johns Hopkins Bloomberg School of Public Health, Baltimore, MD, USA. [10] Department of Biostatistics, Johns Hopkins Bloomberg School of Public Health, Baltimore, MD, USA. [11] These authors contributed equally: Emily E. Burke, Joshua G. Chenoweth. *email: ronald.mckay@libd.org; andrew.jaffe@libd.org

Human induced pluripotent stem cells (hiPSCs) present a unique opportunity to generate and characterize different cell types potentially representative of those in the human brain that may be difficult to ascertain during development or isolate from postmortem tissue. Better-characterizing corticogenesis, and identifying subsequent deficits in psychiatric and neurological disorders, has been the focus of much hiPSC and human embryonic stem cell (hESC) research over the past decade[1]. These cellular models have been especially appealing to study neurodevelopmental disorders where direct analyses of neurodevelopmental changes in patients that would be diagnosed with disorders later in life are very challenging[2].

Transcriptomics is an increasingly powerful tool in both brain and stem cell research. In addition to changing gene expression, alternative RNA splicing is also an important regulator of cortical development[3]. Ongoing large-scale efforts use gene expression data from postmortem human brains to further identify molecular mechanisms that mediate genetic risk for psychiatric disease[4]. Regulation of specific mRNA isoforms associated with genetic risk for psychiatric disease has been modeled in human iPSCs as they differentiate toward neural fates[5]. However, to date, many large-scale transcriptomics efforts in stem cells with respect to corticogenesis have either focused on differentiation of mouse ESCs[6], or single hESC lines profiled in bulk-like CORTECON (WA-09 line)[7] or at the single-cell level like Close et al. (H1 line)[8]. Also, most large-scale hiPSC resources focus on the identity and quality of the initial iPSCs without corresponding RNA-seq data following these iPSCs through differentiation, like Kilpinen et al. (301 donors)[9], iPSCORE (222 donors)[10], and Salomonis et al. (58 iPSC lines)[11], which were all subject to extensive quality control steps[12].

Here we present the results of a hiPSC transcriptomics study on corticogenesis from multiple donors and replicate lines across nine time points (days 2, 4, 6, 9, 15, 21, 49, 63, and 77 in vitro) that represent defined transitions in differentiation: self-renewal, early differentiating cells (accelerated dorsal), neural precursor cells (NPCs), assembled neuroepithelial rosettes[13], and more differentiated neuronal cell types (Table 1). We first identify widespread transcriptional changes occurring across the model of corticogenesis that were largely not influenced by the genetic backgrounds of the iPSCs. We reprocessed and reanalyzed existing hESC- and iPSC-based resources across both bulk and single-cell data in the context of our differentiation signatures and showed similar trajectories of neurogenesis. We then demonstrate in silico that more molecularly mature neuronal cultures are achieved through the addition of rodent astrocytes. We last demonstrate that almost half (48%) of the RNA in our neuronal cultures after 8 weeks of differentiation reflected signatures of adult cortical neurons. We show that these RNA fractions can also be included in downstream analyses to assess or reduce

technical variability and potentially enhance cell-type-specific phenotypic effects. These data, and software tools for barcoding the maturity of iPSC-derived neuronal cells, are available in a user-friendly web browser (http://stemcell.libd.org/scb) that can visualize genes and their transcript features across neuronal differentiation and corticogenesis. We anticipate that these data and our approaches will facilitate the identification and experimental interrogation of transcriptional and post-transcriptional regulators of neurodevelopmental disorders.

## Results

**Differentiating hiPSCs to mature neuronal cultures**. We reprogrammed fibroblasts from the underarm biopsies of 14 donors creating a median 5.5 iPSC lines (interquartile range: 3–7) per subject that were free from karyotyping abnormalities (see the "Methods" section). Fluidigm-based qPCR expression profiling confirmed the loss of FAP (fibroblast activation protein alpha) expression (Supplementary Fig. 1A, $p < 2.2 \times 10^{-16}$) and the gain of *NANOG* expression (Supplementary Fig. 1B, $p = 1.87 \times 10^{-13}$, linear model with random donor effect). We subsequently differentiated iPSCs from five of the donors and fourteen total lines (which had comparable expression versus those donors and subclones not selected) toward a neural stem cell specification, followed by cortical neural progenitor cell (NPC) differentiation and expansion, followed by neural differentiation/maturation (see the "Methods" section). RNA-seq confirmed the expected temporal behavior of canonical marker genes in 13 of the lines, including the loss of pluripotency gene *POU5F1/OCT4* (Fig. 1a), gain of *HES5* (Fig. 1b) through NPC differentiation, and gain of *SLC17A6/VGLUT2* expression through neural maturation (Fig. 1c).

High-content imaging confirmed the self-organization of NPCs into neuroepithelial rosettes[13] (Fig. 1d). Electrophysiological measures taken at 49, 63, and 77 days in vitro (DIV), corresponding to 4, 6, and 8 weeks following NPC expansion, of our neuronal samples cocultured with astrocytes show maturation[14] (Figs. 1e, f). A subset of lines were further interrogated with immunocytochemical labeling of neurons at 8 weeks of differentiation (see the "Methods" section), and showed expected labeling of pre- and postsynaptic proteins (Supplementary Fig. 2). This highlights the ability of our protocol to create neuronal cell lines that display hallmark signatures of neuronal differentiation and are electrophysiologically active.

**Global transcriptional signatures of maturing neural cells**. We first sought to transcriptionally characterize this iPSC model of corticogenesis across five conditions: self-renewal, dorsal fate specification, NPCs, self-organized rosettes, and maturing neural cells. We, therefore, performed stranded total RNA-seq following

### Table 1 Sample and cellular condition information.

| | Accelerated dorsal (days 2, 4, 6, 9) | NPC (day 15) | Rosette (day 21) | Neuron + rat astrocyte (days 49, 63, 77) | Self-renewal (days 2, 4, 6) | Neuron (day 77) | Rat astrocyte (days 49, 63, 77) |
|---|---|---|---|---|---|---|---|
| Donor 3 | 8 | | | | 6 | | |
| Donor 21 | 12 | 4 | 3 | 10 | 3 | 1 | |
| Donor 66 | 10 | 3 | 3 | 7 | 3 | 1 | |
| Donor 90 | 10 | 2 | 1 | 7 | 3 | 1 | |
| Donor 165 | 14 | 3 | 2 | 7 | 3 | 1 | |
| Rat | | | | | | | 3 |
| Total | 54 | 12 | 9 | 31 | 18 | 4 | 3 |

The first four conditions—accelerated dorsal, NPC, rosette, and neuron + rat astrocyte—make up the differentiation time course. Additional cellular conditions used in analyses included self-renewal samples that did not differentiate, purified human neurons, and purified rat astrocytes

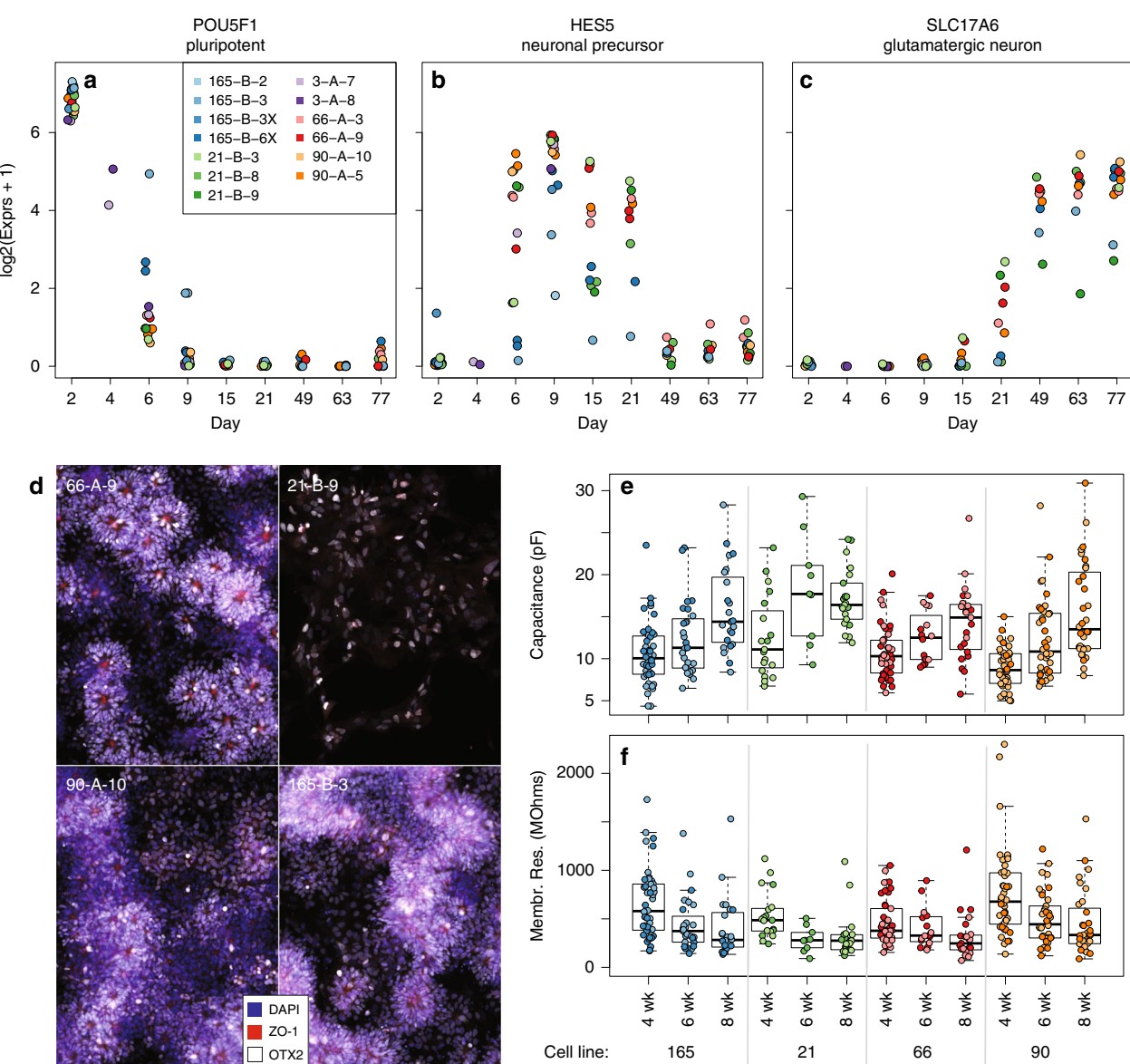

**Fig. 1 Differentiating hiPSCs follow expected trajectories of neuronal development.** Normalized expression levels from RNA-seq showing the expected temporal behavior of canonical marker genes through differentiation: **a** the loss of pluripotency gene *POU5F1/OCT4*, **b** the expression of *HES5* through NPC differentiation, and **c** the gain of *SLC17A6/VGLUT2* through neural maturation. **d** Presence of self-aggregating neural rosettes using representative images from one subclonal line across four donors. Lines clockwise from top left: 66-A-9, 21-B-9, 165-B-3, and 90-A-10. Blue—DAPI; red—ZO-1; white—OTX2. Electrophysiology measurements across neuronal maturation show **e** increasing capacitance and **f** decreasing membrane resistance.

ribosomal depletion on a total of 165 samples, sampling from nine time points across five donors and a series of technical samples (see the "Methods" section). All samples passed batch effect and technical quality control (Supplementary Fig. 3A–C). Six samples were dropped from the cell line that differentiated slower than others (Supplementary Fig. 3D) and five samples were dropped because of identity mismatches (Supplementary Fig. 4).

We first confirmed the representativeness of our iPSC cell lines and subsequent differentiation with the recently published ScoreCard reference data[15] (Supplementary Fig. 5A)—our self-renewal/iPSC lines showed mean 98.1% pluripotency identity (standard deviation (SD) = 1.5%), which significantly decreased through differentiation (Supplementary Fig. 5B, $p < 2.2 \times 10^{-16}$, linear model). We then characterized the global landscape of transcriptional changes accompanying these differentiating and maturing cells using principal component analysis (PCA) of gene

expression levels. The largest component of variability represented neurogenesis, while the second PC further separated those samples in the NPC stage from self-renewing and neuronal cells (Fig. 2a). Both of these components were highly conserved in reprocessed data from the CORTECON hESC time-course dataset (Fig. 2a, Supplementary Fig. 6, PC1 $p = 3.49e–9$, paired one-sided Pearson correlation, PC2 $p = 1.65e–8$, linear regression) even though these cell lines were ESC differentiated, processed, and sequenced in different labs. We further identified global similarity between our differentiating and maturing cells to recently available single- and pooled cell-level data from Close et al.[8], reprocessed using the same pipeline (Supplementary Fig. 7, see the "Methods" section).

We then performed weighted gene co-expression network analysis (WGCNA)[16] to identify more dynamic patterns of expression across neural differentiation and maturation. We identified 11 signed co-expression modules (Fig. 2b) that reflected

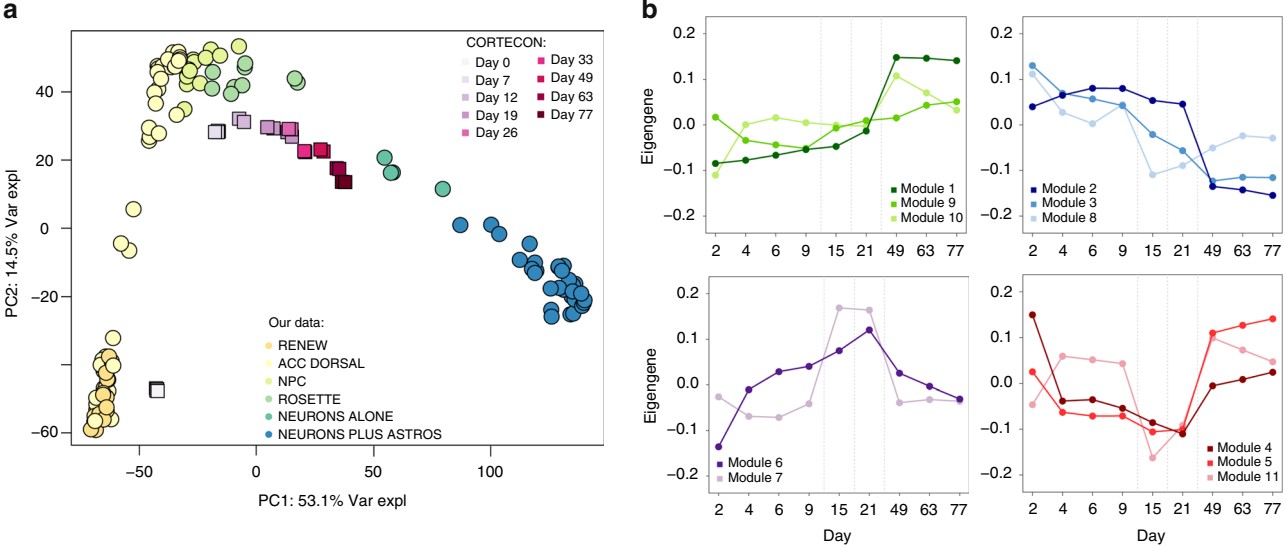

**Fig. 2 Global expression comparison to CORTECON. a** PCA of gene expression levels showing PC1 (53.1% of variance explained) representing corticogenesis and PC2 (14.5% of variance explained) separating samples in the NPC stage from self-renewing and neuronal cells, as well as the conservation of these components of variability in the CORTECON dataset. **b** Eigengenes of the 11 WGCNA modules created from the RNA-seq data of 25,466 expressed genes (with 3284 genes in the unassigned module), grouped by dynamic expression pattern: genes that are more highly expressed in mature neurons, those that are more lowly expressed in mature neurons (related to loss of pluripotency), those that rise in NPCs and then fall, and those that fall in NPCs and then rise again in neurons.

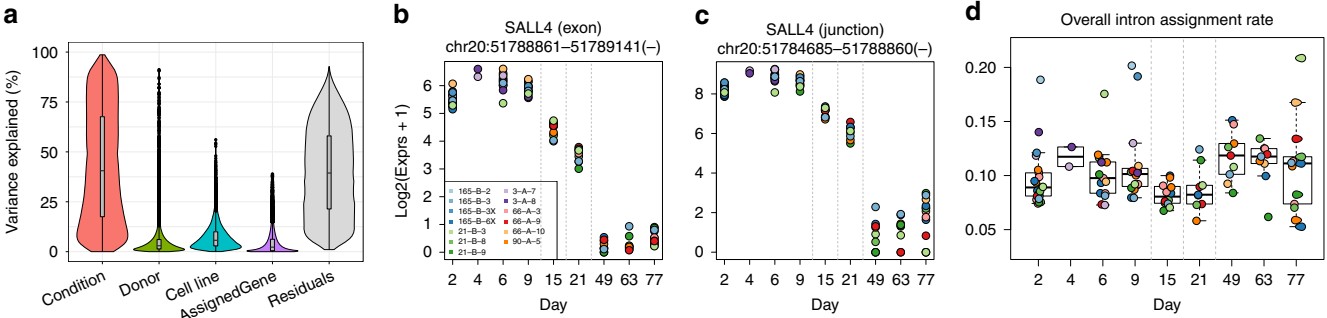

**Fig. 3 Feature-level differential expression. a** The contribution of variance in expression models at the gene level. **b** An example of a differentially expressed exon across conditions from *SALL4*, a gene thought to play a role in the development of motor neurons, as well as expression of the neighboring exon–exon junction (**c**). **d** Percent of aligned reads assigned to intronic sequences across all samples and timepoints, with a significant overall gain in intronic assignment rate between NPCs/rosettes (days 15 and 21) and differentiated neurons (days 49–77) in line with previous research.

known stages of differentiation, which were confirmed using gene set enrichment analysis (Supplementary Data 1), including genes related to loss of pluripotency (Module 3), rise of NPCs (Module 7), and also genes become more highly (Modules 1 and 5) and more lowly (Module 2) expressed in neuronal cultures. We calculated the corresponding eigengenes in these modules in the reprocessed CORTECON dataset, and showed replication for many of the same temporal patterns (Supplementary Fig. 8). These differentiating cells therefore show analogous global expression patterns as other previously published datasets of corticogenesis[7,8].

**Feature-level expression patterns of differentiating neurons.**
We next tested for developmental regulation of individual genes and their transcript features among the 106 time-course samples (excluding 18 self-renewing lines and 4 neuronal lines that were not cultured on rodent astrocytes, Table 1) using statistical modeling (see Methods). By partitioning gene expression variability into different components, we demonstrated that differentiation (condition) explains much more variability in expression than donor/genome or sub-clonal line for the majority

of expressed genes (19,475 genes, 76.4%, Fig. 3a). The majority of expressed genes changed in expression (false discovery rate (FDR) < 0.01) across differentiation and maturation (20,220 genes, 79.4%), including 9067 genes differentially expressed between accelerated dorsal differentiation and NPCs, 1994 genes between NPCs and rosettes, and 12,951 genes between rosettes and neuronal cultures. Across the time course, we also found widespread differential expression at the exon (70.5% of unique genes), exon–exon splice junction (66.2%), and full-length transcript summarizations (72.1%) (Figs. 3b, c, Supplementary Table 1). We further found that a subset of our differentially expressed junctions corresponded to unannotated transcript sequence, including 7298 exon-skipping events and 15,002 alternative exonic boundaries (Supplementary Table 2), a much larger number than that previously reported in smaller studies of hESC differentiation[3]. We have created a user-friendly database to visualize feature-level expression across neuronal differentiation, available at http://stemcell.libd.org/scb.

In an additional analysis related to splicing, we looked into the increase in intron retention (IR), which has previously been

shown to play an important role in differentiation[17]. We identified an overall gain in the percentage of aligned reads assigned to introns between our NPCs/rosettes and differentiated neuronal cells (Fig. 3d, $p = 0.002$, linear model), in line with previous research describing intron retention as a mechanism of rapid gene regulation in response to neuronal activity[18]. In addition, gene ontology (GO) analysis on the genes with significantly (FDR < 0.001) increasing IR ratios through differentiation showed strongest enrichment for neuronal biological process including synaptic signaling (GO:0099536, $p = 9.23e–7$, adjusted $p = 6.9e–4$, hypergeometric test) and neuron development (GO:0048666, $p = 9.87e–7$, adjusted $p = 6.9e–4$) (Supplementary Fig. 9). These analyses confirm the extensive transcriptional changes occurring during neural differentiation and neuronal maturation among individual transcript classes.

**Coculturing NPCs on astrocytes accelerates maturation.** After quantifying expression patterns across the entire time course, we then focused our attention on interpreting the maturation of the neuronal samples. Given the relatively diminished progenitor-like signature of neuronal cells that were cocultured at the NPC stage with rat astrocytes compared with neuronal cells cultured alone (PC2 in Fig. 2a), we sought to more fully characterize the transcriptional effects of astrocyte coculturing. We first demonstrated that we could accurately separate the expression data from human and rat cocultured cells using RNA-seq read alignment, as an in silico RNA sorting technique. We analyzed RNA-seq data from purified rat astrocytes and human neuronal cells and found little cross-species mapping: human neuronal cells alone had low alignment rates to the rat (rn6) genome (mean = 16.6%, SD = 4.5%) and rat astrocytes alone had low alignment rates to the human (hg38) genome (mean = 10.1%, SD = 2.3%) (Supplementary Fig. 10). We next remapped those rodent reads that aligned to hg38 back to rn6 and those human reads that aligned to rn6 back to hg38, and we found two sets of three highly expressed genes that contributed to the majority of reads that mapped across species (Supplementary Table 3). Each of these six genes have either an orthologous gene or significant match with the Basic Local Alignment Search Tool (BLAST) in the other genome.

It is important to note that we found quasi-mapping techniques, e.g., transcript quantification with software like Salmon[19], did not computationally separate the two species due to the shorter lengths of the k-mer-based indices. We confirmed this by comparing the gene counts (aligned using HISAT2) and transcript counts (pseudo-aligned using Salmon) of our five rat astrocyte samples. We found that when aligned to the rn6 genome the total counts per sample were similar between genes and transcripts (gene mean = 43.9M, transcript mean = 43.1M, $p = 0.89$, paired $t$-test), while when aligned to the hg38 genome the transcript counts were an order of magnitude higher (gene mean = 2.2M, transcript mean = 15.1M, $p = 4.4e–5$), implying that the short k-mers are not capable of distinguishing between sequences of the different species. Still, we showed that the expression profiles of human neurons can be computationally separated from rodent astrocytes using standard RNA-seq read alignments, eliminating the need to separate through flow cytometry that can introduce expression changes in RNAs[20].

After establishing that the species alignments were separated in silico, we then compared the human gene expression profiles between four neuronal lines cultured alone and seven of the same lines cocultured with rodent astrocytes at week 8, and identified 3214 genes differentially expressed (at FDR < 0.05) between the two groups (Fig. 4a, Supplementary Data 2). We performed GO analyses on the sets of up- and downregulated differentially expressed genes, and found significant enrichment of genes related to transporter activity, ion channels, and their activity among cocultured neuronal cells plus astrocytes (Fig. 4b, Supplementary Data 3). The upregulated genes were further enriched for being localized in the ion channel complex (GO:0034702, $p = 1.36e–25$, adjusted $p = 6.86e–23$), post synapse (GO:0098794, $p = 1.35e–24$, adjusted $p = 3.4e–22$), and axon (GO:0030424, $p = 1.68e–21$, adjusted $p = 1.65e–19$). These results corroborate previous observations that coculturing with astrocytes produces cultures with more mature neuronal cell types[21].

We next examined the IR ratios of the lines cultured alone and found their IR ratios to be similar to the less mature NPCs/rosettes (p = 0.68, linear model) and lower than those of the neuronal samples cocultured with astrocytes ($p = 0.036$). Last, using electrophysiological measures we showed increased neuronal maturation from weeks 4 to 8 of coculturing with astrocytes, through both increasing capacitance ($p < 2.2e–16$, Fig. 1e) and decreasing membrane resistance ($p = 2.15e–14$, Fig. 1f).

As a secondary analysis, we turned to the rodent astrocytes (quantified against the rat transcriptome) and asked whether coculturing these with human neuronal cells altered their transcriptomes. We found 1329 rodent genes differentially expressed between astrocytes cocultured with neuronal cells compared with astrocytes alone (at FDR < 0.05, Figs. 4c, d), and the genes more highly expressed when cocultured were strongly enriched for neuron projection morphogenesis (GO:0048812, $p = 9.78e–9$, adjusted $p = 1.66e–05$) and axon development (GO:0061564, $p = 5.19e–6$, adjusted $p = 4.4e–3$).

In our cocultured samples, both the human gene expression profiles of the neurons, as well as the rodent expression profiles of the astrocytes, showed enrichment of genes in GO terms related to cellular maturity. These results together suggest increased synergistic maturation of both neuronal cells and astrocytes when cocultured together.

**RNA deconvolution quantifies cortical neuron subpopulations.** Given the maturation reflected in the expression trajectories and physiological activity among the 8-week (77 DIV) neuronal cells, we sought to more fully characterize the underlying cellular composition of these cultures. Previous computational approaches have focused on global analyses determining the most representative time point in brain development for iPSCs and organoids[22]. Here we instead developed a strategy to quantify the fraction of RNAs from ten different developmental, prenatal, and postnatal neural cell types. We reprocessed and jointly interrogated microfluidics-based single-cell RNA-seq datasets from iPSCs and NPCs, fetal quiescent and replicating neurons, and adult neurons, astrocytes, oligodendrocytes, oligodendrocyte progenitors (OPCs), microglia, and endothelial cells[23,24]. These single-cell data were selected to be more comparable to RNA-seq of bulk cells, including the quantification of the entire gene body (rather than 3′ read counting) and fresh brain tissue (rather than frozen tissue that results in ruptured cell membranes and the need for subsequent sequencing of nuclear, rather than total, RNA). We identified a set of 228 genes (Supplementary Data 4) that could transcriptionally distinguish each of these ten cell classes using feature-selection strategies previously described for cellular deconvolution with DNA methylation data[25]. We then standardized the expression to reduce the technical effects across studies (i.e., created Z scores), performed regression on these RNA profiles to estimate the mean RNA levels for each cell class, and implemented the quadratic programming-based approach of Houseman et al.[26] to perform RNA deconvolution (see Methods).

We describe the algorithm in Fig. 5 using data from a subset of genes (to aid visualization). Figure 5a displays the mean

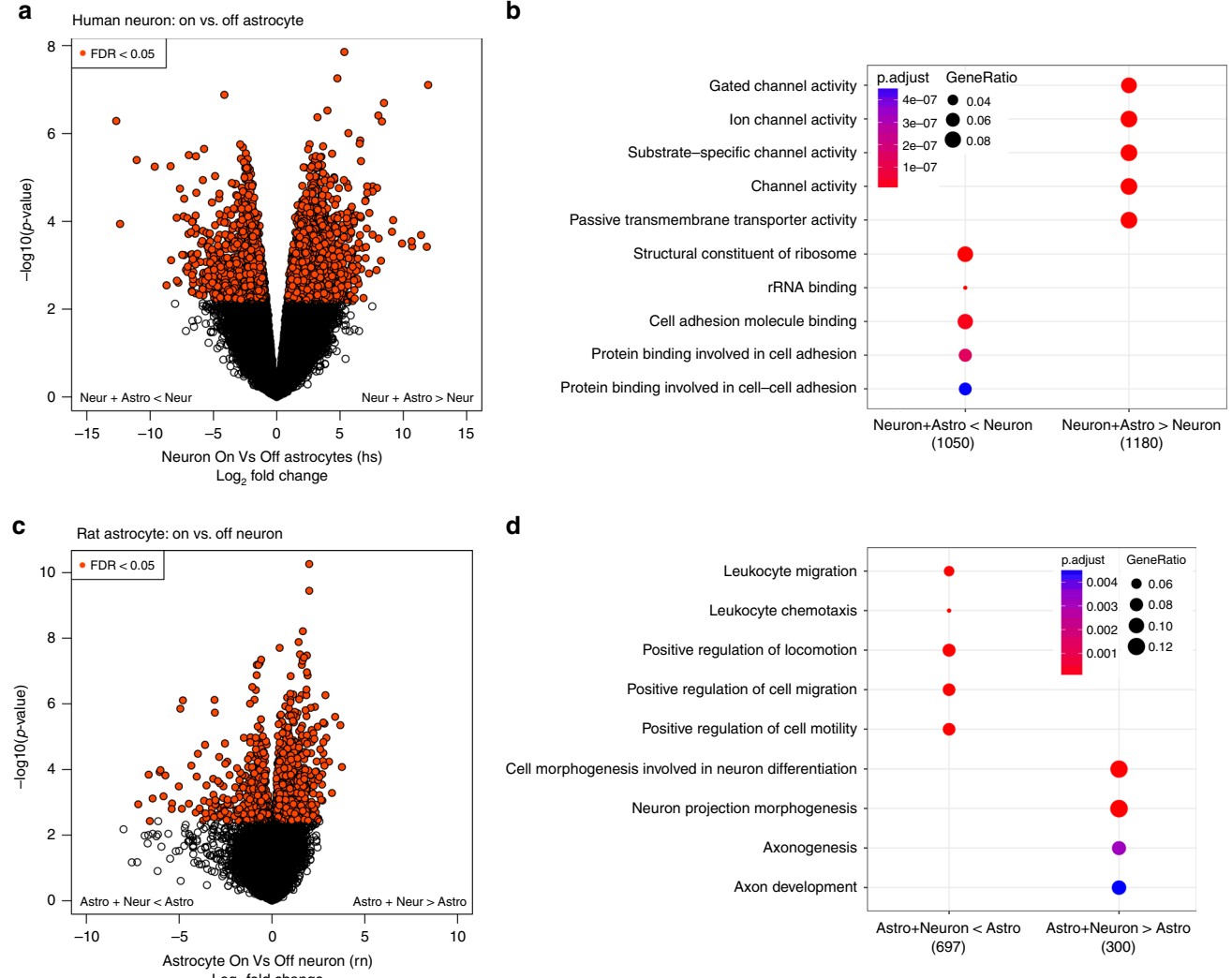

**Fig. 4 Astrocyte coculturing. a** Volcano plot showing the differential expression between human neurons cultured alone and human neurons cocultured with rat astrocytes, as well as the enrichment **b** of the 3214 genes differentially expressed at FDR < 5%, split by direction of change. **c, d** Analogous results of differential expression analysis quantified against the rat genome, comparing rat astrocytes alone with rat astrocytes cocultured with human neurons.

standardized expression levels for each cell class for 131 genes that distinguish six cell classes (vertical lines). Our standardized data across these 131 genes are shown in Fig. 5b, which largely depicts blocks of decreased expression of iPSC and NPC genes, and increased expression of fetal and adult neuronal genes. The expression levels of two adult neuronal genes selected by the algorithm (*SNAP25* and *SCN2A*) across the single-cell reference and our bulk data are shown in Fig. 5c—identical boxplots for all 131 genes can be found in the supplementary materials (Supplementary Figure 11, Supplementary Data 5). We further performed literature searches on the 25 genes that were preferentially expressed in adult neurons used in the RNA deconvolution model and found that these key drivers are being studied throughout the literature, with evidence for mouse-knockout models relating to neurons in 18 of the genes (Supplementary Table 4).

The RNA deconvolution algorithm estimates how similar the expression profile of each sample is to each of the ten reference profiles across these genes, and computes the RNA fraction of each cell class—the shifts of these fractions across our time-course samples are shown in Fig. 5d. More formally, we observed the loss of RNA expression signatures from iPSCs ($p = 1.3e{-}14$), the rise and fall of RNA expression signatures from NPCs (with a similar

pattern as the PC2 of the gene expression data in Fig. 2a), and rise of RNA expression signatures from fetal quiescent ($p = 3.4e{-}33$) and adult ($p = 6.4e{-}32$) neurons (Fig. 5d).

Notably, we also observed significantly increased RNA fractions of adult-like neuronal cells when plated on (48.9%) versus off rodent astrocytes (23.4%, Fig. 5d). The mixture of estimated RNA fractions from neuronal classes from diverse developmental stages highlights the heterogeneity of maturation states from iPSC-derived neuronal cells. However, this approach further demonstrates that a subpopulation of cells in these neuronal cultures are more transcriptionally akin to adult neurons than generally thought, and also provides a computational tool for transcriptionally assessing the relative maturity of differentiated neurons from iPSCs (via the RNA fraction from adult neurons) across independent datasets and experiments.

We do emphasize that this algorithm, when applied to expression data, only estimates the RNA fraction of each cell class, and explicitly not the proportion of cells present in the culture—the exact link between RNA and cellular proportion is unclear. For example, if more mature neurons are larger and more transcriptionally active, they would contain more RNA than other less mature neuronal types.

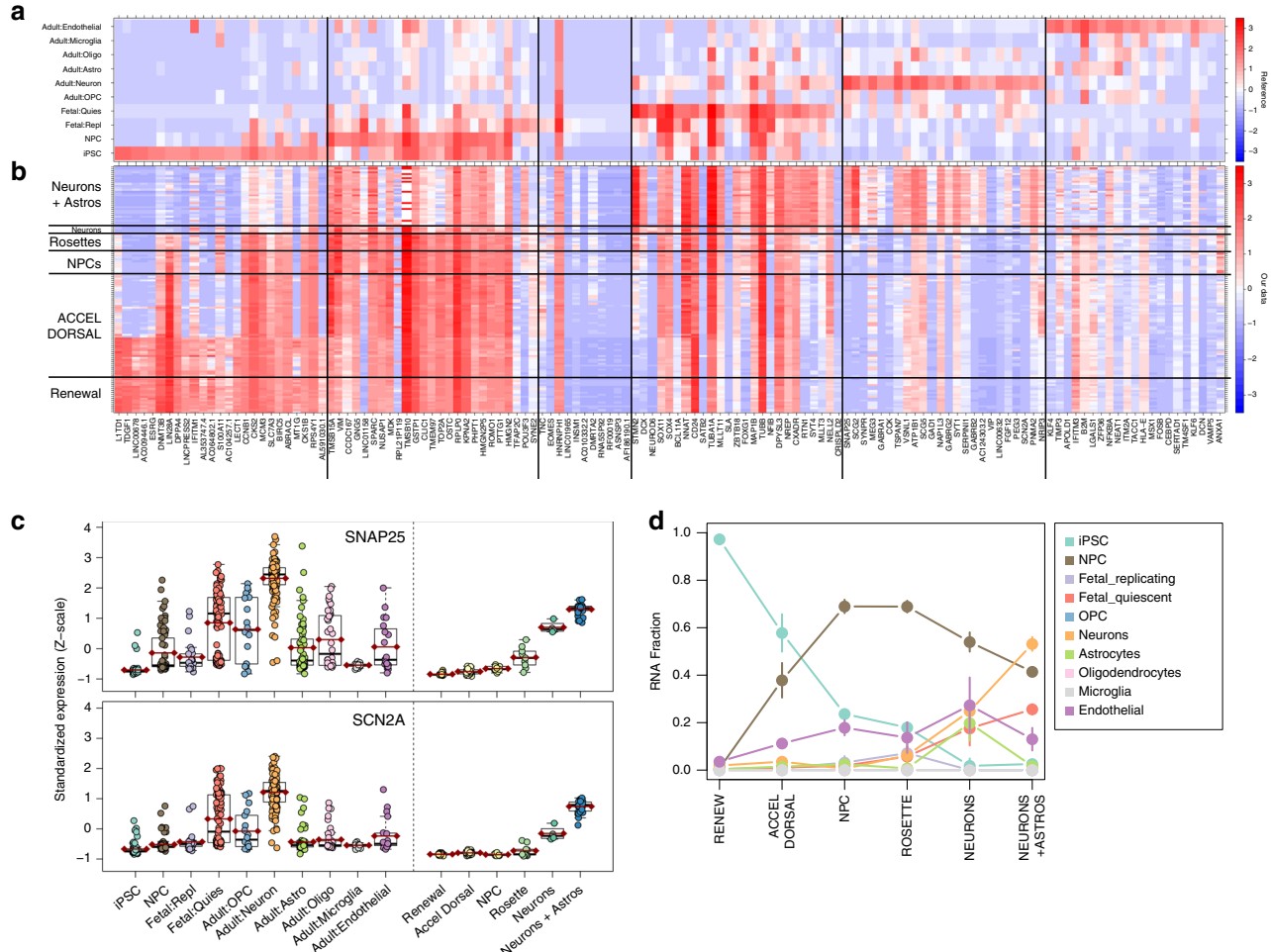

**Fig. 5 Deconvoluting RNA from the underlying cell types across differentiation. a** The mean standardized expression in the single-cell reference data of the 131 genes that were found to distinguish iPSCs (25 genes), NPCs (25 genes), fetal replicating neurons (11 genes), fetal quiescent neurons (25 genes), adult neurons (24 genes), and adult endothelial cells (21 genes). **b** The mean standardized expression of our time-course data across these 131 genes, showing expected higher expression of iPSC genes in the earlier time-course samples, and higher adult neuronal genes in the samples of neurons on astrocytes. **c** Boxplots of the standardized expression in both the single-cell reference data and time-course data of two genes, *SNAP25* and *SCN2A*, that distinguish adult neurons. **d** The RNA fraction of cell types of our bulk data estimated by the deconvolution algorithm, showing the fall of iPSCs and the rise of fetal quiescent and adult neurons.

**Deconvolution: maturation in iPSC-derived neuronal datasets.** To assess the generalizability of our algorithm to samples generated by other labs and research groups, and to confirm the robustness of this deconvolution method to characterize variability in neuronal differentiation protocols, we next turned to a series of published iPSC-derived neuronal and organoid datasets. Within bulk RNA-seq data from differentiating neuronal cells[27], we found increases in the proportion of neurons and astrocytes through differentiation in human iPSCs in a recent schizophrenia and control collection (Fig. 6a). This paper had reported a residual fibroblast-like signature that was not supported by applying this deconvolution to pure fibroblast and iPSC data[15] (Supplementary Fig. 12A). In data from CORTECON[7], we further found a larger proportion of endothelial—that we hypothesize could represent an immature astrocyte signature (see below)—and NPC signatures, and a lack of fetal or adult neuronal classes (Fig. 6b). These less mature cell states—possibly due to the CORTECON protocol not including astrocyte coculturing—were in line with our global PC analysis in Fig. 2a. Similarly, in data from a bulk-developing iPSC dataset in which the authors note that the neurons were harvested during early differentiation stages[28], we found a high proportion of the NPC signature in the neuronal

samples (Fig. 6c). When comparing RNA fractions of bulk data processed across five different labs[29], we found high variability of cell-type proportions by lab, particularly for neurons and endothelial cell types (Fig. 6d). We further examined variability by cell type (see the "Methods" section) and found that the largest amount of variance is from technical variability contributed by lab site for eight of the ten cell-type estimates (Supplementary Table 5). Furthermore, out of the ten cell types, the cell type explaining the largest amount of variance by lab site was the adult neuron RNA fraction ($p = 2.63e{-}13$, nested ANOVA), demonstrating that much of nuisance technical effects represented cellular heterogeneity in maturation.

Analyses in single-cell differentiation datasets further revealed stark differences in the underlying RNA fractions across development, including between DCX– and DCX+ cell populations[8] (Fig. 6e). Other experimental systems and approaches are becoming popular tools in parallel to classical 2D culture conditions, enabling us to make comparisons to 3D cultures such as human organoids. First, we found similar RNA fractions comparing 2D versus 3D directly induced neurons, both on versus off astrocytes, at both 1 and 5 weeks after differentiation[30] (Fig. 6f). Though the differentiation time points between 2D and

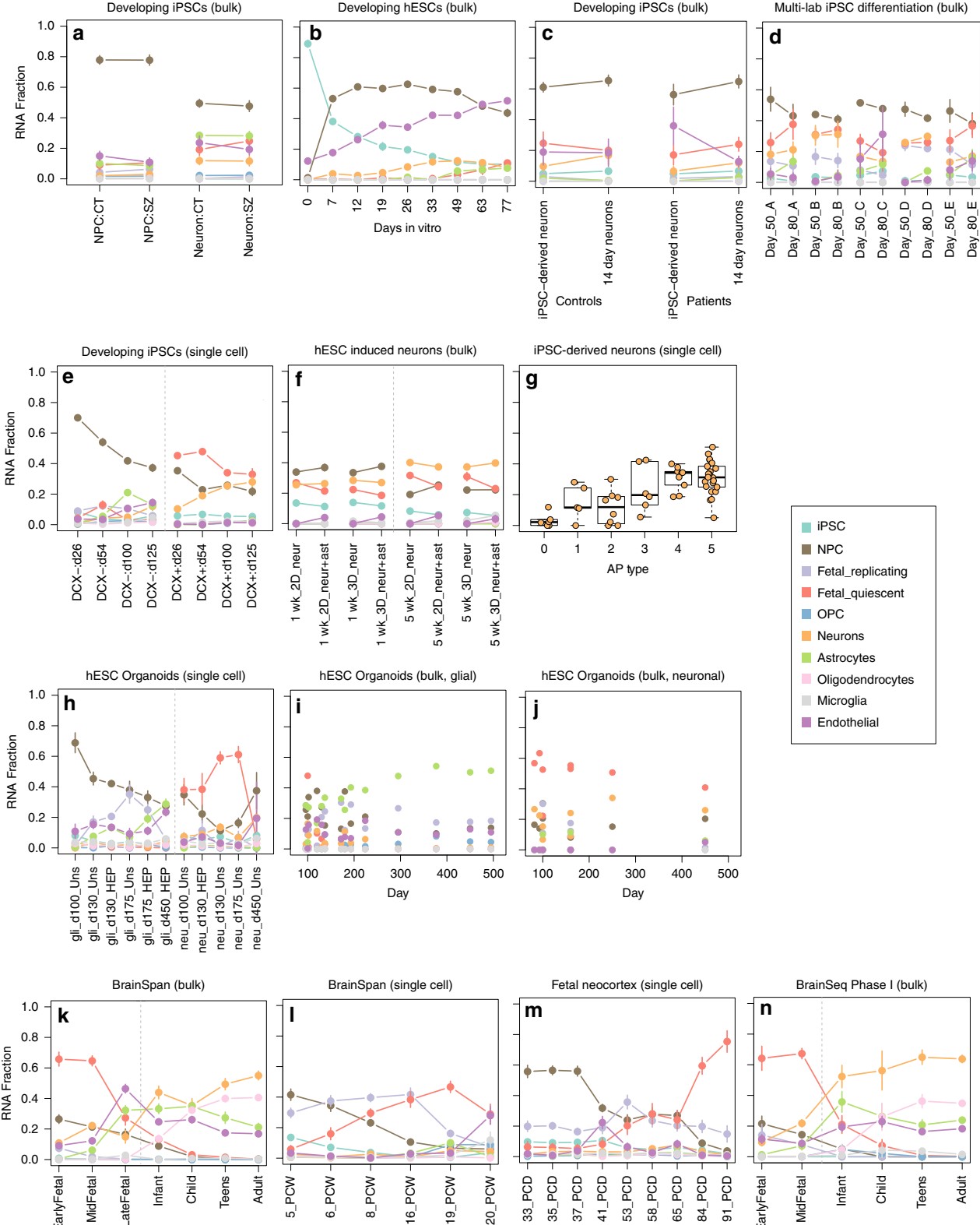

**Fig. 6 Deconvoluting RNA fractions from publicly available RNA-seq data.** We assessed the robustness of our deconvolution algorithm by applying it to various public RNA-seq datasets, including several bulk and single-cell iPSC-derived neuronal (**a**–**g**) and organoid (**h**-**j**) datasets, which included assessing neuronal fractions in a multi-lab study (**d**) and in a Patch-seq dataset with paired electrophysiology data (**g**). We further applied the deconvolution algorithm to multiple studies of human brain tissue (**k**–**n**) comprising fetal through adult samples. Dots represent the mean proportions, while the vertical ticks show standard deviation.

3D protocols may not be directly equivalent, this suggests that standard 2D cultures may generate very similar transcriptionally mature cells to 3D cultures, at least in an induced neuron protocol.

We additionally sought to more functionally validate the RNA fraction originating from the adult neuron class using Patch-seq with paired electrophysiology data[31]. We found a significant positive relationship between the activity states (from 1 to 5, excluding astrocytes, coded as 0) of iPSC-derived neuronal cells and the estimated RNA fraction from neurons (Fig. 6g, $p =$ 3.44e–5, linear model). These data also suggested that the endothelial RNA fraction associated with less mature astrocyte cells (~20% estimate, Supplementary Fig. 12B) is in line with the CORTECON data above.

We also profiled the RNA fractions in iPSC-derived organoids over longer periods of differentiation[32], and found expected RNA distributions within neuronal and glial cellular subtypes, including the rise of RNA from astrocytes in HEPACAM-selected cells (Fig. 6h, i) and RNA from fetal quiescent neurons in neuronal populations (Fig. 6j). We do note that the interpretation of the RNA deconvolution results is somewhat different in the organoid system compared with neuronal cultures, as other cell types besides neurons are intentionally present. These results suggest that this deconvolution algorithm can quantify important contributions to neuronal cellular systems derived from iPSCs, and can be used to remove variability related to differences in differentiation.

**Deconvolution: applicability to brain tissue**. We further assessed the robustness of this RNA deconvolution strategy for assessing neuronal maturity, using a series of publicly available datasets on human brain as a positive control. Using data from the BrainSpan project[33], we confirmed the shift from fetal quiescent to mature adult neurons in homogenate RNA-seq data (Fig. 6k) and the shift from replicating to quiescent fetal neurons in 5–20 postconception weeks (Fig. 6l) and independent 33–91 postconception day (corresponding to 5.5–13 PCW postconception weeks, Fig. 6m) fetal neocortex single-cell RNA-seq datasets.

We performed several additional analyses to independently assess this RNA deconvolution tool. First, we applied this RNA deconvolution approach to a large RNA-seq dataset from postmortem human brain tissue[34], and found largely expected developmentally regulated signals. We observed loss of fetal quiescent neurons ($p < 1$e–100) and NPCs ($p = 6.5$e–91) and the rise of adult neurons ($p = 5.0$e–39), and oligodendrocytes ($p =$ 7.4e–54), primarily at the transition between pre- and postnatal life (Fig. 6n). The estimated RNA fraction of neuronal RNA in the adult samples (mean = 61.9%) was almost twice as high as previous cell count-based approaches, including cytometry-based fractions of NeuN+ cells (mean ~33%)[35] and DNA methylation-based deconvolution of the frontal cortex (27.9%)[36], but was lower than other RNA-based deconvolution strategies applied to DLPFC RNA-seq data (mean = 80%)[37].

**Deconvolution: brain-stage model verifies neuronal signature**. In addition to our cell-type proportion algorithm, we designed and implemented an orthogonal RNA deconvolution model using bulk RNA-seq data from iPSCs and the BrainSpan project to estimate the RNA fractions from eight developmental brain stages (iPSC, early-, mid-, and late-fetal cortex, and infant, child, teen, and adult cortex) using 169 genes (see the "Methods" section, Supplementary Data 6). We first confirmed in the large RNA-seq dataset from postmortem human brain tissue[34] a loss of pluripotency, rise and fall of the early-fetal signature, and then rise of

both mid-fetal and also adult cortical signatures (Supplementary Figure 13A).

In our time-course data, we observed a general loss of pluripotency through differentiation ($p = 3.77$e–38)—the self-renewing and early differentiating cells had high proportions of the iPSC signature (mean 96.1% and 75.1%, respectively), with a rise of early-fetal neocortical-like signature ($p = 4.7$e–7) first in the differentiating cells (20.6%) that became more prevalent in NPCs (33.3%) and rosettes (33.3%) (Supplementary Fig. 13B). We further identified the rise of a mid-fetal neocortical signature ($p = 3.0$e–22) first appearing in rosettes (9.6%) and expanding into neurons off (33.5%) and on (29.9%) astrocytes. The most interesting class switch involved the late-fetal neocortical and adult neocortical classes—neurons grown off astrocytes had high class membership with late-fetal neocortex (10.1%) with low proportions of the adult neocortex (2.7%). However, in the neurons cocultured with rodent astrocytes, these proportions were reversed—these transcriptionally more mature neurons showed that 22.2% of RNA was analogous to the adult neocortex and only 2.0% of RNA reflected late-fetal neocortex. This developmental stage-based deconvolution presents an additional option for deconvolution analyses. In our own data, it further demonstrated the emergence of a more adult-like RNA signature in neuronal cells cocultured on astrocytes.

**Cell-type- and brain-stage-specific models with RNA fractions**. The RNA fractions estimated with our deconvolution methods can further be incorporated directly into statistical models to infer cell-type- or developmental-stage-specific phenotypic effects or technical bias. Specifically the RNA fractions could represent cellular phenotypes in and of themselves, or can be incorporated into downstream analyses, either as quality control checks to ensure comparability of samples or directly into differential expression analysis. To demonstrate the use of these approaches in practice, we completed three additional analyses incorporating cell-type and developmental-stage RNA fractions.

First, we investigated two influencers of neuronal maturation by performing two additional experiments with our iPSC samples across differentiation. We knocked down the expression of *PTEN* and *NRXN1*, two genes implicated in neurodevelopmental disorders, using three complementary short hairpin RNAs (shRNA) for each gene and performing RNA-seq through the rosette stage. We applied our two deconvolution approaches and directly compared the estimated cell-type and brain-stage proportions of our time-course samples, the shRNA control samples, and the two knockdowns (Supplementary Fig. 14A, B) after confirming that the appropriate features were knocked down in expression for *PTEN* (Supplementary Fig. 14C, D) and *NRXN1* (Supplementary Fig. 14E). The deconvolutions showed similar trajectories with little differences in the later stages in both the cell-type model and the brain-stage model—echoing our results of differential expression checks between the time-course and knockdown experiments—suggesting that controlled differentiation of cells produced more comparable cellular cultures, and that these two genes do not alter maturational diversity of NPCs and rosettes. Such a check could show that technical effects do not differ between batches or protocols within a lab before proceeding with analyses.

Next, the RNA fractions estimated with our deconvolution approach could be directly incorporated into differential expression analysis to magnify phenotype effects that might be present in only a subset of cell types. To examine this, we adapted the CellDMC interaction modeling method originally presented for methylation data to one of our deconvoluted RNA-seq datasets[38]. Reanalysis of NPC data from Hoffman et al. (2017)[27] identified

78 genes differentially expressed by schizophrenia diagnosis using the interaction modeling strategy, none of which were found significant using standard DE modeling (Supplementary Fig. 15, Supplementary Data 7, 8). Utilizing these RNA fractions could allow one to find cell-type-dependent differences that could be missed when samples show variable cellular phenotypes.

Finally, the RNA fractions could be used as a check if technical variabilities are consistent across cell type. To demonstrate, we conducted differential expression analysis using the Volpato et al.[29] data consisting of samples across five lab sites. As previously mentioned, we found high variability of cell-type proportions by lab. In a differential expression model with lab and adult neuronal fraction (the most variable cell type by lab), we found 7616 genes to be differentially expressed across the sites; however, when using the CellDMC interaction method we found 1835 DE genes by lab and 1099 DE genes by lab–neuron interaction (bonf < 0.05). Including neuronal proportion interaction greatly lessened the lab effects, demonstrating that incorporating the RNA fractions into models could reduce potential technical effects present in datasets.

All combined, these results from diverse RNA-seq datasets from a variety of experimental approaches demonstrate the wide applicability of this transcriptional deconvolution strategy. Our RNA fraction approach can be a robust tool for a variety of analytical methods, including quantifying technical versus biological effects, finding cell-type-specific expression patterns within phenotypes, and evaluating the cellular maturity of model systems.

## Discussion

Here we extensively characterized the transcriptomes of human iPS cells as they traverse distinct neurodevelopmental transitions defined by morphological and functional features. We used 13 independent cell lines derived from five donors to understand the relevance of these cellular systems to model human brain development. We identified widespread transcript-specific changes in expression across differentiation and neuronal maturation at varying scales among co-expressed genes, and among individual transcript features that were assessed for replication in independent neuronal differentiation datasets. It will be important to relate these data to transcriptional isoforms that associate with risk for neurodevelopmental disorders to build faithful in vitro models for experimental applications. With these data we have also created a web browser visualizing expression levels across neuronal differentiation, which will be a valuable resource for experimental designs, such as transcript-specific knockdown and overexpression experiments.

We focused on the ability to assess neuronal maturity in our paradigm. This was approached both experimentally through use of rodent astrocyte cocultures, and computationally through the use of two related deconvolution implementations to determine the brain stage and cellular identities of differentiating cells that we applied across thousands of samples and cells. While previous reports have demonstrated electrophysiological and transcriptional evidence for the enhancement of neuronal maturity due to coculturing NPCs with astrocytes during differentiation[39–41], to our knowledge few studies have generated cell-type-specific gene expression profiles without the use of cell sorting. Here we show that the functional consequences of coculturing human NPCs with rodent astrocytes can be detected in silico, without the need for cell sorting, by leveraging the mappability of longer sequencing reads. These cross-species analyses revealed the synergistic effect of coculturing on neuron and astrocyte growth, simultaneously promoting the maturation of both the astrocytes and the human neurons.

We have also leveraged tools previously developed for estimating cell-type composition profiles from DNA methylation data to benchmark the developmental and cellular landscapes of human iPSCs as they differentiate toward neural fates. We show that 8-week (77 DIV) neuronal cultures, particularly those cocultured on rodent astrocytes, are a diverse mixture of cells at varying maturities and cellular states. While the majority of cellular and developmental classes are consistent with the prenatal brain as previously reported using microarray-based profiling[22], we have identified subsets of more mature cells also present in our cultures that model later developmental stages. These cells presumably account for those with evidence of more mature phenotypes in our electrophysiology assays. The computational deconvolution tool we have developed here, and validated across several public RNA-seq datasets, can be adopted by researchers to assess the maturation of potentially heterogeneous cultures of iPSC-derived neurons.

In addition, the two reference profiles and subsequent regression calibration-based tools for deconvoluting the relative RNA contributions of cell stages and classes can be easily utilized by researchers using RNA-seq data to ensure more comparable case and control lines for discovering molecular phenotypes. This approach will also be increasingly valuable to assess organoids and complex three-dimensional stem cell models under development[42]. Furthermore, we provided a statistical framework showing that these estimated RNA fractions can be applied in downstream analyses, such as to draw cell-type-specific inference of differential expression analysis or to reduce technical biases related to differentiation within bulk data.

Recent advances in single-cell analysis are being leveraged to comprehensively define the human central nervous system. The BRAIN Initiative has effectively used single-cell transcriptomics and epigenetics to characterize cellular populations in the developing brain[43]. We anticipate that the tools developed in our study will complement those efforts that depend on cellular dissociation and selection that largely restrict data to gene-level and nuclear expression. Future development of mathematical and computational approaches to relate datasets and enhance sparse cell-level insight will be valuable toward understanding brain health and disease. These expression-based resources of neuronal differentiation can provide beneficial metrics for quality control (via deconvolution) and can assist in experimental design (via our expression browser) to more fully leverage the power of cellular systems to better understand and model debilitating brain disorders.

## Methods

**Clinical fibroblasts**. Skin fibroblasts, taken in a superficial circular incision (3 mm in diameter) in the mesial aspect of the upper arm, were cultured after informed consent from neurotypical volunteer subjects who were participants in the Sibling Study of Schizophrenia at the National Institute of Mental Health in the Clinical Brain Disorders Branch (NIMH, protocol 95M0150, NCT00001486, Annual Report number: ZIA MH002942053, DRW PI) with additional support from the Clinical Translational Neuroscience Branch, NIMH (KFB PI). All subjects were extensively screened with obtaining medical, psychiatric and neurological histories, physical examinations, MRI scans, and genome-wide genotyping to rule out diagnosable clinical disorders.

**iPS cell line derivation and culture**. Human iPS cell lines were generated using the Stemgent mRNA reprogramming kit (00-0071) and the Stemgent microRNA Booster kit (00-0073) with modifications[5]. Briefly, human fibroblasts were seeded with 50,000 cells in individual wells of a six-well plate coated with Matrigel in DMEM media + 10% FBS and 2 mM L-glutamine. The next day (day 1), the media was changed with Pluriton human NUFF conditioned media with 300 ng/ml B18R protein. On days 1 and 5, the microRNA booster kit was used with the StemFect RNA transfection reagent kit from Stemgent to enhance reprogramming. On days 2–12, the OSKML RNAs were transfected. The mRNA reprogramming process was performed in a 37 °C, 5% $O_2$, and $CO_2$ incubator. Individual colonies were picked and expanded on irradiated mouse embryonic fibroblasts in DMEM-F12

(Invitrogen), 20% knockout serum replacement (KSR), 5 ng ml⁻¹ FGF2 (R&D Systems), 0.1 mM 2-mercaptoethanol (Sigma), 2 mM L-glutamine, and 1× non-essential amino acids (both from Invitrogen). Putative iPS cell lines were subjected to karyotype analysis (Cell Line Genetics) and molecular analysis prior to the generation of feeder-free working cell banks. Confirmation of known pluripotency genes and silencing of fibroblast-enriched genes[44] in reprogrammed cells was measured using the Fluidigm BioMark System and TaqMan probes according to the manufacturer's protocol. Only cell lines with a normal chromosomal complement were chosen to generate cell banks for this study. For maintenance of banked hiPSCs in feeder-free conditions, cells were dissociated to single-cell populations with accutase (A11105, Life Technologies), plated at a density of $1 \times 10^6$ cells in a Matrigel (BD)-coated six-well plate, and cultured with mTeSR1 (Stem Cell Technologies, #05850)[45]. The cells were cultured with 5 mM Y27632, ROCK inhibitor (Y0503, Sigma-Aldrich) to increase the single-cell survival upon dissociation. At 24 h after plating, Y27632 was removed from the medium, and cells were cultured for another 4 days before the next passaging. Initially, a total of 71 subclonal lines from 13 subjects were reprogrammed (Supplementary Fig. 1). Out of the 48 genes tested by the targeted Fluidigm qPCR there was only a single gene different between the five donors selected for sequencing in this study compared with the other eight donors not selected—which was *GSTT1*, a strong copy number variant with influence on gene expression[46]. There were further no differences in any gene's expression comparing the 10 (out of 13) subclonal lines with RNA-seq data compared with the 24 with just Fluidigm qPCR expression data across these five donors. We therefore believe that the five donors and 13 subclonal lines ultimately selected for RNA-seq were a representative sample.

**Neural differentiation.** To induce neural differentiation, iPSCs were plated under feeder-free conditions described above. Twenty-four hours after plating, media was changed to either the "Dorsal" condition (mTESR1 plus 100 nM LDN193198 and 2 μM SB431542) or the "Accelerated Dorsal" condition (Stem Cell Technologies AggreWell medium plus 100 nM LDN193189 and 2 μM SB431542). Forty-eight hours later (Day 2), the Dorsal condition media was replaced and the "Accelerated Dorsal" condition media was changed to N2/B27 medium plus 100 nM LDN193189 plus 2 μM SB431542. On Days 4 and 6, media for both conditions was replaced. Total RNA was collected (RNeasy Qiagen) under both conditions at Days 2 and 6, and at Day 4 for a subset of samples. The Accelerated Dorsal Condition was exclusively used for continued neural differentiation and media was replaced daily up until Day 9. On Day 9, total RNA was collected, or cells were passaged using Accutase onto Poly-L-ornithine/fibronectin-coated tissue culture dishes in N2 media plus 20 ng/ml FGF2. Media was changed each day up until Day 15 when RNA was harvested, or cells were passaged with HBSS onto Poly-L-ornithine/fibronectin-coated tissue culture dishes in N2 media plus 20 ng/ml FGF2. Media was exchanged each day with fresh N2 until Day 21 when neural rosettes appear. On Day 21, total RNA was collected, or cells were passaged with HBSS onto PDL/Laminin coated coverglass with or without rat E18 astrocytes in N2 media in a humidified 37 °C tissue culture incubator at 5% oxygen. After 24 h, the media was exchanged for Neuronal Differentiation Media (NeuroBasal Invitrogen 12348-017, 1× GlutaMax Invitrogen 35050-061, 3 nM Selenite, 25 μg/ml Insulin Sigma I6634, 1× Pen/Strep final 1× Invitrogen 15140-122, 1× B27 Invitrogen 17504-044, concentration, 10 ng/ml BDNF R&D systems, 248-BD/CF, and 10 ng/ml NT3 R&D systems, 267-N3/CF). In all, 50% of the media was changed every other day until Day 28. After 7 days on astrocytes, 100% of the media was changed to Neuronal Differentiation Media plus 20 μM AraC. In all, 50% of the media was changed with Neuronal Differentiation Media plus 20 μM AraC every other day up to Day 35. At Day 35, 100% of the media was exchanged for Neuronal Differentiation Media without AraC and 50% of media was exchanged every other day for up to 8 weeks. RNA was harvested at indicated intervals during the process.

**Electrophysiology.** Human neuronal cultures on glass coverslips were submerged in our recording chamber and constantly perfused with an external bath solution consisting of (in mM) 128 NaCl, 30 glucose, 25 HEPES, 5 KCl, 2 CaCl₂, and 1 MgCl₂ adjusted to pH 7.35 with NaOH. All recordings were performed at approximately 32 °C. Patch pipettes were fabricated from borosilicate glass (N51A, King Precision Glass, Inc.) to a resistance of 2–5 MΩ. For voltage-clamp measurements, cells were held at –70 mV and recording pipettes were filled with (in mM) 125 potassium gluconate, 10 HEPES, 4 Mg-ATP, 0.3 Na-GTP, 0.1 EGTA, and 10 phosophocreatine, 0.05%, adjusted to pH 7.3 with KOH. Current signals were recorded with either an Axopatch 200B (Molecular Devices) or a Multiclamp700A amplifier (Molecular Devices) and were filtered at 2 kHz using a built-in Bessel filter and digitized at 10 kHz. Data were acquired using Axograph on a Dell PC (Windows 7). We tested for differences in capacitance and membrane resistance across time in culture using linear mixed effects modeling, treating line and donor as random intercepts, and naturally log-transforming these two measures to improve normality assumptions of these models.

**Cellular imaging.** Neural progenitor cells spontaneously self-organize into rosette structures reflecting morphological properties of the developing neural tube[47]. Neural progenitors were plated in 24-well ibidi plates in triplicate for each line. After differentiation for 6 days, an acellular lumen is detectable with antibody

directed against ZO-1. Surrounding the lumen are laminae of dorsal forebrain progenitors identified with anti-OTX2 antibodies. Nuclei were labeled with DAPI. Under these conditions, the structures are largely two-dimensional but are becoming pseudo-3D. An array of 6 × 6 wide-field (non-confocal) images was captured automatically from each of three neighboring wells using the Operetta and processed using custom code in Columbus (both Perkin Elmer). Briefly, ZO-1 segmentation was used to reveal the core of each rosette. Anti-OTX2 immunoreactivity was used to identify the limits of each rosette. Nuclear morphology was measured indirectly using the DAPI signal.

For the neural staining, iPSC-derived neural monolayers were cultured on 24-mm glass coverslips. Samples were fixed with ice-cold 4% paraformaldehyde for 15 min. Permeabilization and blocking were performed simultaneously with 10% normal goat serum containing 0.1% Triton X-100 for 30 min. Primary antibodies directed against BIII-Tubulin (TuJ1), PSD-95, and synapsin1 (Syn1) were prepared in PBS containing 10% normal goat serum and incubated with sample overnight at 4 °C. Fluorescently conjugated Goat-anti-x, y, z secondary antibodies were prepared in 10% normal goat serum and incubated with sample for 2 h at room temperature. Coverslips were mounted in medium containing DAPI. All samples were imaged on a Zeiss 780 LSM microscope with 63 × 1.4 NA objective across a range of Z positions at optimal step size for 3D reconstruction. In total, 3 × 3 adjacent fields were stitched to create example images. Ten or more of these 3 × 3 arrays were captured per coverslip.

**RNA sequencing.** Total RNA was extracted from samples using the RNeasy Plus Mini Kit (Qiagen). Paired-end strand-specific sequencing libraries were prepared from 300 ng of total RNA using the TruSeq Stranded Total RNA Library Preparation kit with Ribo-Zero Gold ribosomal RNA depletion (Illumina). An equivalent amount of synthetic External RNA Controls Consortium (ERCC) RNA Mix 1 (Thermo Fisher Scientific) was spiked into each sample for quality control purposes. The libraries were sequenced on an Illumina HiSeq 3000 at the LIBD Sequencing Facility, after which the Illumina Real Time Analysis (RTA) module was used to perform image analysis and base calling, and the BCL converter (CASAVA v1.8.2) was used to generate sequence reads, producing a mean of 58.3 million 100-bp paired-end reads per sample.

**RNA-seq: processing pipeline.** Raw sequencing reads were mapped to the hg38/GRCh38 human reference genome with splice-aware aligner HISAT2 version 2.0.4[48]. Samples without rodent tissue averaged 86.6% alignment rate (SD = 5.1%), while 31 samples cocultured with rat astrocytes averaged 33.9% alignment rate (SD = 10.1%) to the human genome. Feature-level quantification based on GENCODE release 25 (GRCh38.p7) annotation was run on aligned reads using featureCounts (subread version 1.5.0-p3)[49] with a mean 62.9% (SD = 8.2%) of mapped reads assigned to genes for human samples, and a mean 43.1% (SD = 9.4%) of mapped reads assigned to genes for samples containing rat astrocytes. Exon–exon junction counts were extracted from the BAM files using regtools[50] v. 0.1.0 and the 'bed_to_juncs' program from TopHat2[51] to retain the number of supporting reads (in addition to returning the coordinates of the spliced sequence, rather than the maximum fragment range) as described in ref. [34]. Annotated transcripts were quantified with Salmon version 0.7.2[19] and the synthetic ERCC transcripts were quantified with Kallisto version 0.43.0[52]. For an additional QC check of sample labeling, variant calling on 740 common missense SNVs was performed on each sample using bcftools version 1.2. After processing, statistical analyses were completed using R versions 3.3 and 3.4, and combined figures were generated with Adobe Illustrator version 22.0.1.

**RNA-seq: quality control.** After preprocessing, samples were checked for quality control measures. All samples passed quality control checks for alignment rate, gene assignment rate, mitochondrial mapping rates, and ERCC spike-in concentrations. We looked for batch effects by checking for differences in technical metrics by batch, or for separation by batch within top principal components. In addition, we clustered the gene expression of three sets of replicates sequenced on seven different flow cells, finding that the replicates clustered by sample and not flow cell. The replicates were included for quality control purposes and were not used in any additional analyses. All of our batch effect checks indicated consistency across batch. Next we examined expression through the differentiation time course of known pluripotency and neuronal differentiation marker genes to confirm that our cell lines differentiated as expected; one cell line (six samples) did not pass this marker check due to slow differentiation and was dropped from all analyses. Finally, a genotype check was conducted to confirm the donor labeling of all samples. Called coding variants were matched to existing microarray-based genotype calls, and five samples were dropped due to ambiguous sample identity.

**RNA-seq: rat astrocytes.** Purified rat astrocytes from five samples, as well as the cocultured neuron/astrocyte samples and four samples of human neurons, were processed with an analogous rat pipeline as discussed above (see Processing pipeline). The samples were aligned to the rn6/Rnor_6.0 genome, and feature counts were quantified using the Ensembl release 86 annotation of the rat transcriptome. A mean 90.7% (SD = 2.1%) of rat astrocyte reads aligned to the reference genome and cocultured samples had an average 68.4% (SD = 9.9%) alignment

rate, while human neurons averaged 16.7% (SD = 4.5%) alignment. The expression levels of the three groups were compared to each other in assessing the neuronal maturation effects of coculturing with astrocytes.

**Public data processing**. Multiple public human RNA-seq datasets were downloaded and used to quantify the developmental stage and cellular composition of our samples across the differentiation time course. All public data below were run through the same processing pipeline as outlined above to get comparable read counts and expression values (see Processing pipeline).

Raw FASTQ files from each of the following datasets were downloaded from the sequencing read archive (SRA).

(1) CORTECON (van de Leemput et al.[7]) included 24 single-end (50 bp) samples of human cerebral cortex development from hESCs across nine timepoints (days 0, 7, 12, 19, 26, 33, 49, 63, and 77) [SRP041179, GSE56796].

(2) The ScoreCard dataset (Choi et al.[53], Tsankov et al.[15]) consisted of 73 paired-end (2 × 100 bp) samples of hESCs, hiPSCs, and fibroblasts [SRP063867, GSE73211].

(3) Song et al.[23] consisted of 174 single-cell samples of iPSCs, NPCs, and motor neurons from both paired- and single-end libraries [SRP082522, GSE85908].

(4) Close et al.[8] included 1733 single-cell and 40 pooled samples of paired-end (2 × 50 bp), hESC-derived cortical interneurons at four timepoints of differentiation (days 26, 54, 100, and 125) profiling both neurons (DCX+) and progenitors (DCX−) [SRP096727, GSE93593].

(5) Darmanis et al.[24] included 420 paired-end (2 × 75 bp) single-cell samples of four embryonic and eight adult brains from eight types of cortical tissue [SRP057196, GSE67835].

(6) Bardy et al.[31] consisted of 56 paired-end (2 × 100 bp) single-cell samples of iPSC-derived neurons, as well as the corresponding Patch-seq electrophysiology data.

(7) Tekin et al.[30] included 157 bulk paired-end reads of induced neurons both on and off human and mouse astrocytes at 1 and 5 weeks after differentiation.

(8) Volpato et al.[29] included 57 paired-end samples of bulk tissue processed across five different labs and two cell lines at days 50 and 80 of differentiation.

(9) Lin et al.[28] consisted of 19 paired-end (2 × 100 bp) samples in early differentiating human neurons derived from iPSCs of 22q11.2 DS schizophrenia patients and controls.

In addition, FASTQ files from both bulk and single-cell RNA-seq datasets were downloaded from the BrainSpan atlas[33].

(10) Homogenate data consisted of 407 single-end (75 bp) samples from neocortical regions of 41 donors aged early fetal through adult (40 year old).

(11) Single-cell data were 932 single-end (100 bp) samples from the DLPFC and dorsal pallium of eight early- to mid-fetal brains.

The following data from the BrainSeq Phase I consortium were downloaded and reprocessed[34]:

(12) The BrainSeq data consisted of one DLPFC sample, each from 318 unique donors, ranging from fetal through age 85, with paired-end (2 × 100 bp) libraries.

Other datasets used for evaluating the cellular deconvolution approaches included:

(13) Raw FASTQ files from human organoids (both bulk and single-cell data) from Sloan et al.[32], which were processed with the above pipeline [SRR5676732].

(14) Publicly available gene counts (not FASTQ files) for Ensembl v70 from Hoffman et al.[27] for 94 iPSC-derived NPC and neuronal samples.

**Statistical analysis: WGCNA**. To identify dynamic patterns of gene expression across neuron maturation, we performed signed weighted gene co-expression network analysis (WGCNA)[16] using the software's R package. The analysis was performed on 25,466 expressed genes (cutoff mean RPKM > 0.1) from 106 samples across all time points (days 2, 4, 6, 9, 15, 21, 49, 63, and 77). Self-renewal samples and neurons cultured without rat astrocytes were not included in the WGCNA analysis. Normalized expression values in the form of $\log_2(RPKM+1)$ were used, with gene assignment rate (as a measure of sample quality) regressed out of the expression matrix. To find clusters, the software first selected a soft thresholding power of six, then, using a minimum module size of 30 and maximum block size of 10,000, assigned 22,182 of the expressed genes to 11 signed modules representing dynamic expression patterns through differentiation. GO analysis was then carried out on the modules to find biological processes and functions enriched by the gene sets of each cluster. To evaluate replication of the expression patterns, we separated the reprocessed CORTECON data into the same 11 gene sets, and compared both the eigengenes and GO terms of the 11 modules calculated in each of the two datasets.

**Statistical analysis: time-course DE**. Again looking at 106 samples and 25,466 expressed genes, we performed differential expression analysis to find genes with changing expression through the stages of neuronal differentiation. Our statistical model used the voom method[54] of linear modeling to estimate the mean-variance relationship of the gene log counts, adjusting for cell line and the proportion of mapped reads assigned to genes (gene assignment rate). We found genes most differentially expressed between each of the neighboring cell conditions of early neuronal differentiation, NPC, rosettes, and neurons. Within the same model we

also found genes differentially expressed across the entire differentiation time course. Voom modeling was run on genes, exons, exon–exon junctions, and annotated transcripts, with p values adjusted for false discovery rate (FDR) within feature type.

To find the prevalence of unannotated junctions in public datasets, we used the snaptron_query() function from recount2[55,56]. We queried the 49,657 samples in the Sequence Read Archive (SRAv2) aligned to hg38 with a different aligner (Rail-RNA). A feature was considered present in Snaptron if it was found in over 1% of samples (>497).

**Statistical analysis: alternative splicing**. Alternative splicing events were further investigated through intron retention (IR) measures. Ratios of retained introns to spliced introns were calculated using IRFinder version 1.1.1[57], and linear regression models were used to obtain a list of genes with significantly increasing or decreasing IR ratios across the time course, adjusting for cell line and gene assignment rate. GO analysis on the directional gene lists was then completed. In addition, the percent of aligned reads assigned to introns was calculated by featureCounts for each sample using a GTF of the intronic features of GENCODE release 25. Differences in intron assignment rate by condition were evaluated with a linear regression model adjusting for cell line.

**Statistical analysis: astrocyte effects**. To assess the effect of astrocytes on neuronal maturation, we compared the gene expression at day 77 of four neuronal lines cultured alone with seven of the same lines cocultured with rodent astrocytes, for 24,706 genes with average expression over 0.1 RPKM. A voom model adjusting for cell line and gene assignment rate was implemented to find genes significantly differentially expressed between the two groups at FDR < 0.05. We then performed GO analysis on the upregulated and downregulated gene sets to investigate enrichment of the DE genes. Similarly, using a voom model adjusting for day and gene assignment rate, we tested for differential expression of 17,908 expressed rat genes between the cocultured samples and three purified rat astrocytes on days 49, 63, and 77, followed by GO analysis on the up- and downregulated genes.

**RNA deconvolution modeling**. We used two regression calibration models to determine the relative compositions of our iPSC model system. The first model involved identifying the relative developmental stage of our sequenced cells, using data from the ScoreCard[15,53] and BrainSpan[33] homogenate sequencing projects. The second model involved identifying the relative cellular composition of our sequenced cells, using the Fluidigm-based single-cell RNA-seq datasets from Song et al.[23] and Darmanis et al.[24].

**RNA fraction model**. We combined single-cell normalized expression data $(\log_2(RPKM+1))$ from 63 iPSC and 73 NPC samples from Song et al.[23], and 25 replicating and 110 quiescent fetal neurons, 18 oligodendrocyte progenitor cells (OPCs), 131 neurons, 62 astrocytes, 38 oligodendrocytes, 16 microglia, and 20 endothelial cells from Darmanis et al.[24]. We defined cell-type-specific genes using the same framework described by Jaffe and Irizarry 2014[25], which involved creating a "barcode" of 25 genes per cell type that were more highly expressed for one cell type compared with all others (t-statistic p-value < 1e−15), and we ranked by $\log_2$ fold changes for selection. From our final set of 228 unique genes, we scaled each gene expression value to the standard normal distribution to improve comparability between single-cell and bulk RNA-seq data, and created the regression calibration design matrix based on Houseman et al.[26], shown in Supplementary Data 4. We again then projected samples into the design matrix using the 'projectCellType()' function in the *minfi* Bioconductor package.

**Developmental-stage model**. We combined normalized expression data $(\log_2(RPKM+1))$ from the 21 iPSC samples in the ScoreCard project and 407 neocortical bulk samples from the BrainSpan project across seven timepoints (73 early-, 73 mid-, and 17 late-prenatal, as well as 53 infant, 71 child, 55 teen, and 65 adult postnatal). We defined stage-specific genes using the same framework as the cell proportion model. Here we created a "barcode" of 25 genes per stage, which were more highly expressed for each stage compared with all others (t-statistic p-value < 1e−15), and ranking subsequent significant genes by $\log_2$ fold changes for selection. As some stages had similar expression levels with others, we ended up with 169 unique genes and created the regression calibration design matrix based on Houseman et al.[26], shown in Supplementary Data 6. We then projected samples into this design matrix using the 'projectCellType()' function in the *minfi* Bioconductor package[58]. Forty-four genes were shared between these two statistical models for deconvolution.

**Volpato et al.[29] data analysis**. We assessed variability with cell line, day in vitro, and lab site after first running the RNA cell-type deconvolution algorithm to get RNA fraction estimates. Then, for each of the ten cell types, we ran an ANOVA with the full model (RNA fraction ~cell line + DIV + lab) and a nested model leaving out one of the three covariates. The p-values for three ANOVAs for each cell type were recorded as shown in Supplementary Table 5, quantifying the amount each covariate contributes to the model for that cell type.

**Reporting summary**. Further information on research design is available in the Nature Research Reporting Summary linked to this article.

## Data availability

Links for downloading all sequencing reads, including both the time-course data and all reprocessed public data, are available at http://stemcell.libd.org/scb/. Transcriptional data are deposited under accession code PRJNA596331. Source data are available as a Source Data file.

## Code availability

Accompanying processing and analysis code is available at https://github.com/lieberinstitute/libd_stem_timecourse.

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

## Acknowledgements

The authors gratefully acknowledge Lieber Institute for Brain Development and the IMED Biotech Unit of AstraZeneca for funding and support of this work.

## Author contributions

E.B. performed data analysis and interpretation and wrote the paper. J.C. contributed to the design of the study, performed cellular reprograming, quality control, and experiments, and wrote the paper. J.S. contributed to the design of the study and generated RNA-sequencing data. L.C.T. performed data analysis and interpretation. S.K. contributed to the design of the study, analysis, and performed cellular differentiation. N.M. contributed to the design of the study, analysis, and performed cellular differentiation. Y.W. performed cellular reprogramming, differentiation, and experiments. C.C. contributed to the design of the study. R.S contributed to the design of the study and data interpretation. D.Ho. contributed to the design of the study, performed experiments and cellular imaging, and interpreted data. H.C. performed physiology experiments. A.S. performed cellular reprograming and differentiation, quality control, and experiments. K.S. performed cellular reprograming. G.H. performed physiology experiments. M.B. contributed to neuronal cell culture and performed physiology experiments. B.P. analyzed data. W.U. created the searchable database and browser. C.V. provided data analysis, A.J. performed cellular differentiation. A.P. contributed to data analysis and interpretation. A.R. generated RNA-sequencing data. S.S. performed data analysis. R.B. contributed to the design of the study and data interpretation. J.B. contributed to the design of the study and the data interpretation. D.Hi. contributed to cellular reprogramming. S.P performed cellular physiology experiments. K.M. contributed to the design of the study and data interpretation. T.H. contributed to the design of the study and the interpretation. J.K. contributed to the design of the study and the interpretation. K.B. contributed to the clinical research study that generated source fibroblasts. J.A. contributed to the clinical research study that generated source fibroblasts. A.C. contributed to the design of the study and the interpretation. N.B. contributed to the design of the study and the interpretation. D.W. contributed to the design of the study and the interpretation, contributed to the clinical research study that generated source fibroblasts, and wrote the paper. B.M. contributed to the design of the study and the interpretation, performed cellular physiology experiments, and wrote the paper. R.M. contributed to the design of the study and the interpretation. A.E.J. contributed to the design of the study and the interpretation, performed data analysis, and wrote the paper.

## Competing interests

R.W.B., A.J.C. and N.J.B. were full-time employees and shareholders of AstraZeneca at the time these studies were conducted. No other authors have competing interests.
