## [Peer Review File · Nature Communications]

Reviewers' comments:

Reviewer #1 (Remarks to the Author):

In their revised manuscript entitled "Dissecting transcriptomic signatures of neuronal differentiation and maturation using iPSCs," Burke, Chenoweth et al present analysis generated from forebrain neuron differentiation of 5 hiPSC donors (13 subclonal lines) across nine time points broadly capturing self-renewal, early neuronal differentiation, NPCs, assembled rosettes, and functional neurons, which is now validated across both a larger set of 13 donors (71 clonal lines) and 48 genes as well as a series of published datasets. Their differentiation and maturation signature represents a valuable dataset and analytic method for the hiPSC community, one that I hope will be incorporated into many reports moving forward once published in Nature Communications.

Technically, the analyses are quite solid, although the authors could improve their manuscript by more fully demonstrating across a much larger number datasets and analyses exactly how much power to resolve disease and genotype specific effects is gained by applying these new analyses. It would be great if the authors could better clarify the applicability of their signatures and methods more broadly to the field. I hope that in doing so, this might better address the major concerns of all three reviewers.

The authors used Hoffman et al as a proof-of-concept dataset to identify 78 genes previously undetected with significant main effects of diagnosis. Does this teach us anything more broadly about hiPSC models? Schizophrenia? If the authors could repeat this across more existing datasets (the Lin et al samples may have been too immature, but surely there are dozen(s) of alternative datasets to try), perhaps the authors could say more about the types of biology (do they better connect to disease risk genes?) and/or types of genes (ie. synaptic?) their methods better capture? Is their signature specific to forebrain differentiation protocols or can it also be used, for example, to clarify hiPSC-based motor neuron studies of ALS or applied to NGN2-based inductions? Overall, the impact of this manuscript would be increased if the authors (even) better demonstrated the utility, generalizability and biological relevance of their analyses.

Reviewer #2 (Remarks to the Author):

Overall, we think the paper shows a great effort of analysing five different iPSC lines during their neuronal differentiation, we find the co-culturing experiment interesting and we think they show a good analysis of publicly available databases. However, the way the paper is currently written is hard to understand and does not have a clear message. As it is now, the paper is probably more suited for a more specialised journal. To guide you through our decision please find below a more detailed analysis of the Reviewer #2 and #3 comments:

Reviewer #2

1) We think considering that the authors were starting from five lines, going up to analyse 13 lines is a great effort. However, they still only differentiated five lines which doesn't provide a suitably large dataset. Furthermore, the table that they present is confusing if the journal decides to publish this paper it needs to be clarified.

2)The authors didn't fully answer the reviewer's comment, which we think was very valid. It would

have been interesting to pick the most significant differentially expressed splice variant and analyse the proteins that derive from them.

3) This is a very valid comment. We think it is still interesting that they see a transcriptional change between neurons differentiated with or without rat astrocytes and it is possible that it is a rough example of what happens with human astrocytes. So overall I think it is still informative.

4) The reviewer's point about the cross comparison between 2D and 3D cultures is valid. While the authors seem to run this analysis as proof of principle, we feel that this section is not as informative. Furthermore, we think that the paper does not have a very coherent line and that every section seems to be disjointed from the previous one.

Reviewer #3

1) Although the authors extended the sample size they did not differentiate them into neurons or astrocytes so the information about differentiation is unchanged.

2) We agree with the reviewer in saying that the paper does not bring a lot of novelty to the field. However, we do find the co-culturing experiment very interesting and they do develop a useful tool.

Reviewer #1 (Remarks to the Author):

In their revised manuscript entitled “Dissecting transcriptomic signatures of neuronal differentiation and maturation using iPSCs,” Burke, Chenoweth et al present analysis generated from forebrain neuron differentiation of 5 hiPSC donors (13 subclonal lines) across nine time points broadly capturing self-renewal, early neuronal differentiation, NPCs, assembled rosettes, and functional neurons, which is now validated across both a larger set of 13 donors (71 clonal lines) and 48 genes as well as a series of published datasets. Their differentiation and maturation signature represents a valuable dataset and analytic method for the hiPSC community, one that I hope will be incorporated into many reports moving forward once published in Nature Communications.

Technically, the analyses are quite solid, although the authors could improve their manuscript by more fully demonstrating across a much larger number datasets and analyses exactly how much power to resolve disease and genotype specific effects is gained by applying these new analyses. It would be great if the authors could better clarify the applicability of their signatures and methods more broadly to the field. I hope that in doing so, this might better address the major concerns of all three reviewers.

The authors used Hoffman et al as a proof-of-concept dataset to identify 78 genes previously undetected with significant main effects of diagnosis. Does this teach us anything more broadly about hiPSC models? Schizophrenia? If the authors could repeat this across more existing datasets (the Lin et al samples may have been too immature, but surely there are dozen(s) of alternative datasets to try), perhaps the authors could say more about the types of biology (do they better connect to disease risk genes?) and/or types of genes (ie. synaptic?) their methods better capture? Is their signature specific to forebrain differentiation protocols or can it also be used, for example, to clarify hiPSC-based motor neuron studies of ALS or applied to NGN2-based inductions? Overall, the impact of this manuscript would be increased if the authors (even) better demonstrated the utility, generalizability and biological relevance of their analyses.

We appreciate the positive comments on our analyses and the usefulness of the resource. We have updated the manuscript to better clarify the applicability of our deconvolution signatures. To better show the utility of cell type-specific analyses, we have also included a third analysis example – using the Volpato et al. dataset with differentiation across five lab sites – to understand how differential expression by lab site differences are affected by including neuronal cell type proportions. We found far fewer DE genes associated with lab sites in a model with lab and neuronal fraction interactions, showing that including the RNA fraction in the dataset can reduce potential technical effects as well. This analysis including additional details has been added to the revised manuscript.

Reviewer #2 (Remarks to the Author):

Overall, we think the paper shows a great effort of analysing five different iPSC lines during their neuronal differentiation, we find the co-culturing experiment interesting and we think they show a good analysis of publicly available databases. However, the way the paper is currently written is hard to understand and does not have a clear message. As it is now, the paper is probably more suited for a more specialised journal. To guide you through our decision please find below a more detailed analysis of the Reviewer #2 and #3 comments:

Thank you for your comments. We have added edits to improve the paper’s message and to better illuminate the novelty and usefulness of our resource. We have emphasized what we believe are the most important findings of this manuscript, which can hopefully refine some of the more detailed comments below:

Result #1: We have created an expression browser across neuronal differentiation, which we believe will be important for experimental design, including designing transcript-specific knockdown and overexpression experiments.

Result #2: We have demonstrated that co-culturing human neurons with rodent astrocytes promotes maturation of the neurons, and their transcription profiles can be isolated with RNA-seq in silico without the need for cell sorting

Result #3: We have created a computational tool to assess the maturation of potentially heterogeneous cultures of iPSC-neurons which we validate across many public RNA-seq datasets.

Result #4: We have provided a statistical framework for using these estimates of neuronal maturation to draw cell type-specific inference of differential expression analysis and reduce technical biases related to differentiation.

Reviewer #2

1) We think considering that the authors were starting from five lines, going up to analyse 13 lines is a great effort. However, they still only differentiated five lines which doesn't provide a suitably large dataset. Furthermore, the table that they present is confusing if the journal decides to publish this paper it needs to be clarified.

The authors agree that at this point it is not feasible to increase the number of donors. As mentioned and expanded on in our previous response, we believe these five donors and 13 subclonal lines are a representative sample, and that the number of subclonal lines we use show appropriate replication and support the results we have drawn. Table 1 has been updated and is now presented in a clearer manner.

In the context of our four main results above, this potentially limited sample size only affects Result #1, and not the other 3 main results. We further note that the expression profiles from the multiple subclonal lines from these 5 donors show very similar patterns across differentiation (Figure 3A), and these data clearly had adequate statistical power to identify differences in expression across differentiation – for example, almost every expressed gene is differentially expressed from iPSCs to neurons. Obviously, more subtle phenotypes would require a much larger sample size, but it is unclear how much more useful our expression browser would be with the inclusion of additional donors since the effect of differentiation is so substantial.

2) The authors didn't fully answer the reviewer's comment, which we think was very valid. It would have been interesting to pick the most significant differentially expressed splice variant and analyse the proteins that derive from them.

We did appreciate this comment from the reviewer – however, in relation to our 4 main findings above, the lack of protein validation of the previously unannotated splicing only would detract from a small portion of Result #1, since we believe the vast majority of researchers will use our data and browser to interrogate expressed sequences. Taking the reviewers' comments into account, we have therefore summarized the results of the unannotated splice junctions into a single sentence in the first paragraph of the "Feature-level expression patterns of differentiating neural cells" results subsection.

However, we did expend considerable time and effort attempting to validate some of these novel isoforms to address this reviewer's suggestion, and no immediately clear candidate was found, given the current technological limitations of protein assays and antibody specificity. First, we took the list of the top 573 differentially expressed exon-skipping junctions across the time-course ($p < 2e-15$). We used the following criteria to consider a splice variant a potential candidate:

1. Expressed in our iPSC samples (mean RPKM>0.04 in accelerated dorsal samples). We reasoned that iPSCs would be the most viable cell type for this work, as they are more easily grown and are strictly human cells (since many protein antibodies are cross species and we co-culture our neurons with rodent astrocytes).
2. Has only one or two known protein isoforms. We reasoned that more complex transcript structures and presumable protein isoforms would be harder to achieve clearly discernable western blot gels.
3. Expressed in >1% of Snaptron samples (which are annotated in Table S3). Here we reasoned that unannotated splice junctions that have already appeared in published RNA-seq datasets in other cell types and tissue sources are more likely to be real, particularly when this resource used an independent RNA-seq read aligner (Rail-RNA versus HISAT2 here).
4. Present & expressed in at least half of the RiboTag mouse model samples from Baser et al. (2019) dataset [PMID: 30700908]. We reasoned that RNAs present on the ribosomes in RiboTag models of adult neural stem cells would be much more likely to a) exist in human neural stem cells and b) validate at the protein level since the RNAs are presumably actively being translated on the ribosomes. Interestingly 65% of these 573 unannotated splice junctions were present in this mouse dataset after performing liftOver to the mm10 genome, and 190 were completely novel in mouse (whereas at least one end of the junction was annotated to a known exon in human).
5. Simple transcript structure with a small number of exons.

After applying these filters, there were only 7 potential junctions remaining. Most junctions were filtered by steps 2 and 4 (few protein isoforms and RiboTag expression).

We further used the Biostrings and seqinr packages in R to predict the molecular weights of both the annotated transcript, and the novel transcript with the exon(s) spliced out. We first found that one of our 7 remaining junctions were not found in the CDS. We further found for the remaining candidates that the changes in molecular weights were very small, suggesting that protein qualification and quantification approaches like western blots would not be able to uniquely identify the novel variant even if it was translated into a protein.

We then went back and found what proportion of junctions were in the CDS, and what the predicted change in molecular weight would be for three groups of junctions: 1) the top 573 ($p < 2e-15$), 2) the 7,298 DE junctions with $FDR < 0.01$, and 3) all 18,303 DE exon-skipping junctions. We found that, consistently across all 3 groups, roughly 1/3 of the junctions are not in the CDS, and only roughly 10% would have a change in molecular weight greater than 10 kilodaltons.

Finally, we noticed a promising junction in our top 573 with a large change in molecular weight due to a large exon being spliced out (ANK2: chr4:113,346,023–113,360,822(+)), reducing from approximately 430 kDa to 200 kDa. However, after further investigation we observed that two of ANK2's other annotated transcripts were also weighted around 200 kDa, again making the transcript with novel splice variant indistinguishable on a western blot.

This initial analysis did result in creation of an in-house R function that takes a list of either annotated or custom transcripts and returns their predicted molecular weights. Overall, we found no clear candidate junction to continue forward with proteomics validation, and the authors believe that additional proteomics work is outside the scope of this study. Because we did not complete further validation of these variants, we instead have deemphasized the unannotated splicing section of the manuscript into a single sentence as mentioned above.

3) This is a very valid comment. We think it is still interesting that they see a transcriptional change between neurons differentiated with or without rat astrocytes and it is possible that it is a rough example of what happens with human astrocytes. So overall I think it is still informative.

Thank you for your comments.

4) The reviewer's point about the cross comparison between 2D and 3D cultures is valid. While the authors seem to run this analysis as proof of principle, we feel that this section is not as informative. Furthermore, we think that the paper does not have a very coherent line and that every section seems to be disjointed from the previous one.

The authors have added an additional sentence within our organoid discussion to make our intentions clearer. "We do note that the interpretation of the RNA deconvolution results is somewhat different in the organoid system compared to neuronal cultures, as other cell types besides neurons are intentionally present."

We have also completed minor edits throughout the paper to give the paper a better flow between sections.

Reviewer #3

1) Although the authors extended the sample size they did not differentiate them into neurons or astrocytes so the information about differentiation is unchanged.

At this point in time it would be difficult to add differentiated samples to our analysis. As discussed in our previous response, we explained the sampling strategy, and we believe our smaller discovery sample set appropriately supports our findings, and in particular, motivated the reanalysis of over 5000 publicly available samples that act as supplementary examples and support for our methods. We further highlight that this potential weakness only affects Result #1 above, and not the other three main results.

2) We agree with the reviewer in saying that the paper does not bring a lot of novelty to the field. However, we do find the co-culturing experiment very interesting and they do develop a useful tool.

Thank you for your remark on our co-culturing experiment. Additionally we have added further edits to better illuminate the novelty and usefulness of our resource, and we have better highlighted the four main results in the Discussion section of the manuscript.

REVIEWERS' COMMENTS:

Reviewer #2 (Remarks to the Author):

We believe that the authors have shown a great effort in modifying the manuscript according to the reviewers' comments and that the paper has majorly benefited from the authors edits. We think that the manuscript could now be suitable for publication in Nature Communication however, we suggest some minor edits:

- 1) Fig1E and 1F are hard to read. Please clarify the cell lines analysed in the figures.
- 2) The database developed by the authors (<http://stemcell.libd.org/scb>) has the potential to be an extremely useful tool however, it is hard to navigate. Please make it more user friendly by specifying more details on the exons and jxn sections. In addition to a better generalized explanation specific examples of use would be appropriate.

Overall, the manuscript has greatly improved and we are sure it will make a meaningful contribution to the field.

REVIEWERS' COMMENTS:

Reviewer #2 (Remarks to the Author):

We believe that the authors have shown a great effort in modifying the manuscript according to the reviewers' comments and that the paper has majorly benefited from the authors edits. We think that the manuscript could now be suitable for publication in Nature Communication however, we suggest some minor edits:

- 1) Fig1E and 1F are hard to read. Please clarify the cell lines analysed in the figures.
- 2) The database developed by the authors (<http://stemcell.libd.org/scb>) has the potential to be an extremely useful tool however, it is hard to navigate. Please make it more user friendly by specifying more details on the exons and jxn sections. In addition to a better generalized explanation specific examples of use would be appropriate.

Overall, the manuscript has greatly improved and we are sure it will make a meaningful contribution to the field.

We appreciate your positive feedback and additional editing suggestions. We have simplified the axis labels in Figures 1E and 1F to be cleaner to read. Additionally we have extensively updated the database resource to include a better introduction and instructions and a clearer layout, making it overall more user friendly.